# Synthesis and Photocatalytic Activity of TiO_2_/CdS Nanocomposites with Co-Exposed Anatase Highly Reactive Facets

**DOI:** 10.3390/molecules26196031

**Published:** 2021-10-04

**Authors:** Yi-en Du, Xianjun Niu, Xinru He, Kai Hou, Huiling Liu, Caifeng Zhang

**Affiliations:** 1Department of Chemistry and Chemical Engineering, Jinzhong University, Jinzhong 030619, China; xjniu1984@163.com (X.N.); hexr123456@163.com (X.H.); houk456@163.com (K.H.); 2Department of Mechanics, Jinzhong University, Jinzhong 030619, China; 3Department of Chemistry, Taiyuan Normal University, Jinzhong 030619, China

**Keywords:** TiO_2_/CdS nanocomposites, highly reactive facets, synergistic effects, photocatalytic activity

## Abstract

In this work, TiO_2_/CdS nanocomposites with co-exposed {101}/[111]-facets (NH4F-TiO2/CdS), {101}/{010} facets (FMA-TiO2/CdS), and {101}/{010}/[111]-facets (HF-TiO2/CdS and Urea-TiO2/CdS) were successfully synthesized through a one-pot solvothermal method by using [Ti_4_O_9_]^2−^ colloidal solution containing CdS crystals as the precursor. The crystal structure, morphology, specific surface area, pore size distribution, separation, and recombination of photogenerated electrons/holes of the TiO_2_/CdS nanocomposites were characterized. The photocatalytic activity and cycling performance of the TiO_2_/CdS nanocomposites were also investigated. The results showed that as-prepared FMA-TiO_2_/CdS with co-exposed {101}/{010} facets exhibited the highest photocatalytic activity in the process of photocatalytic degradation of methyl orange (MO), and its degradation efficiency was 88.4%. The rate constants of FMA-TiO_2_/CdS was 0.0167 min^−1^, which was 55.7, 4.0, 3.7, 3.5, 3.3, and 1.9 times of No catalyst, CdS, HF-TiO_2_/CdS, NH_4_F-TiO_2_/CdS, CM-TiO_2_, Urea-TiO_2_/CdS, respectively. The highest photocatalytic activity of FMA-TiO_2_/CdS could be attributed to the synergistic effects of the largest surface energy, co-exposed {101}/{010} facets, the lowest photoluminescence intensity, lower charge-transfer resistance, and a higher charge-transfer efficiency.

## 1. Introduction

Since Fujishima and Honda discovered that titanium dioxide (TiO_2_) could photodecompose water to produce hydrogen in the 1970s [1], TiO_2_, as a traditional semiconductor material, has been widely utilized in the fields of photocatalytic degradation of pollutants, dye-sensitized solar cells, lithium-ion batteries, gas sensors, etc., due to its relatively good chemical stability, low cost, non-toxicity, and environmentally friendly nature [2,3,4]. However, TiO_2_ is subject to many limitations in industrial application due to its relatively large bandgap (anatase: ~3.2 eV, rutile: ~3.0 eV) and the relatively rapid recombination rate of photogenerated electrons and holes [5,6]. Recent studies have shown that changing the crystal phase, grain size, morphology, specific surface area, heterogeneous structure, and exposed facets of TiO_2_ is an effective way to increase the photocatalytic activity. Especially, the configuration of heterogeneous structure with exposed highly reactive facets plays an important role in improving the charge separation efficiency and photocatalytic activity of TiO_2_ [7]. Therefore, it is necessary to design and synthesize TiO_2_-based semiconductor composite photocatalyst with exposed highly reactive facets to broaden its optical light absorption range and accelerate the separation of photogenerated electrons and holes, thus improving the photocatalytic efficiency [8,9]. To extend the light absorption range of TiO_2_ to the visible region, numerous efforts have been made to combine TiO_2_ with other semiconductors, such as Fe_2_O_3_ [10], CdS [11], Cu_2_O [12], Ag_3_PO_4_ [13], WO_3_ [14], ZnS [15], and Bi_2_S_3_ [16]. Among various inorganic semiconductors, cadmium sulfide (CdS) is considered to be one of the best cocatalyst candidates because of its appropriate bandgap (2.4 eV), low price and appropriate band edge position for photocatalytic redox reaction [4,17]. Therefore, TiO_2_/CdS composites have been widely applied in the fields of photocatalysis and photovoltaics due to their simple composition, easily controlled microstructure and high extinction coefficient [18,19]. For example, Gao et al. reported the synthesis, characterization, and photocatalytic activity of the CdS-loaded TiO_2_ microspheres with exposed {001} facets [20]. Wang et al. demonstrated that coating CdS nanoparticles on anatase TiO_2_ films with exposed {001} facets can greatly improve the photoelectrochemical water splitting ability [21]. Dai et al. prepared surface-fluorinated anatase TiO_2_ nanosheets with exposed {001} facets/CdS-diethylenetriamine nanobelts composites, which improved the ability of water to split into H_2_ under visible light [22]. Ma et al. prepared RGO/CdS/TiO_2_ nanocomposite with high-energy {001} TiO_2_ facets, which showed high photocatalytic activity and stability under visible light [23]. The different crystal surfaces of anatase TiO_2_ in the TiO_2_/CdS composites, such as {101} (0.44 J/m^2^), {010} (0.53 J/m^2^), {001} (0.90 J/m^2^), and {111} (1.61 J/m^2^) facets, are determined by the different surface energies [24,25]. Using appropriate morphology-controlling agents can selectively adsorb on the surface of TiO_2_ crystal with more active sites, which can not only reduce its surface free energy, but also inhibit the crystal growth along the corresponding direction, so as to expose specific crystal facets [26]. Among the various types of capping agents, fluorine morphology-controlling agents, such as hydrofluoric acid (HF), ammonium fluoride (NH_4_F), and non-fluorine morphology-controlling agents, such as urea, and formic acid (FA) are often employed. Generally speaking, the Fˉ ionized by HF and NH_4_F are easy to adsorb on the {001}-faceted surfaces of anatase TiO_2_ to reduce its surface energy and facilitate its growth [27,28]. However, it is undeniable that HF and NH_4_F will generate toxic and corrosive substances at high temperature [29], and it is difficult to remove the passivation of Fˉ on the crystal surfaces of anatase TiO_2_ [30]. Owing to these limitations, a novel methodology for synthesizing anatase TiO_2_ crystals with exposed specific surfaces using morphology-controlling agents without fluorine was developed. Carbon dioxide decomposed by urea at high temperature is converted into carbonate in alkaline solution, which can adsorb on the {001} crystal planes of anatase TiO_2_ and reduce its surface energy, thus leading to the formation of {001} crystal planes [29]. As an organic weak acid, formic acid is preferentially attached onto the specific {101} surfaces to reduce its surface energy and facilitate the oriented growth of crystal along [1] direction during the growth of anatase TiO_2_ crystal, resulting in the reduction of {001} crystal plane [31].

In this study, TiO_2_/CdS nanocomposites with co-exposed anatase {101}/[111]-facets, {101}/{010} facets and {101}/{010}/[111]-facets were prepared using different morphology-controlling agents by a simple hydrothermal treatment of [Ti_4_O_9_]^2−^ colloidal solution containing CdS crystals. The photocatalytic degradation of MO performance of the as-obtained TiO_2_/CdS nanocomposites was investigated under ultraviolet irradiation. To our knowledge, this is the first report that TiO_2_/CdS nanocomposites with co-exposed anatase highly reactive facets exhibit excellent photocatalytic activity for MO degradation.

## 2. Materials and Methods

### 2.1. Synthesis of CdS Nanoparticles

The synthesis method of CdS nanoparticles was as follows: 7.1190 g of Cd(NO_3_)_2_·4H_2_O and 1.9036 g of thiourea were added to a hydrothermal vessel. Then 30 mL of deionized water and 30 mL of absolute ethyl alcohol were added into the above vessel and magnetically stirred for 30 min. Then, 1.0080 g of ammonium fluoride (NH_4_F) was added to the above-mixed solution and continue stirring for 30 min. After that, the above autoclave with a white solution was sealed and put into a constant temperature blast drying oven (GZX-GF101-1-BS, Hefei Kejing Material Technology Co. Ltd., Hefei, Anhui, China) and maintained at 180 °C for 24 h. Finally, the orange–yellow precipitates were collected by centrifugation and washed several times with deionized water.

### 2.2. Preparation of [Ti_4_O_9_]^2−^ Colloidal Solution from H_2_Ti_4_O_9_

Layered potassium tetratitanate fiber K_2_Ti_4_O_9_ was synthesized via conventional solid-state calcination using potassium carbonate (K_2_CO_3_) and titanium dioxide (TiO_2_) as raw materials [32]. Briefly, 14.5115 g K_2_CO_3_ (0.105 mol) and 31.9681 g TiO_2_ (0.400 mol) were ground evenly in an agate mortar and then heated in a high-temperature box resistance furnace at 900 °C for 24 h using an alumina crucible. The resultant K_2_Ti_4_O_9_ (30.00 g) was immersed in 1.0 mol/L HNO_3_ aqueous solution (3 L) under stirring conditions for 3 days to complete the K^+^/H^+^ exchange, during which the HNO_3_ aqueous solution was replaced with a fresh one every day to prepare the H_2_Ti_4_O_9_. 8.00 g H_2_Ti_4_O_9_, 2.50 g tetramethylammonium hydroxide (N(CH_3_)_4_OH·5H_2_O), and 80 mL of deionized water were added to a hydrothermal vessel and stirred magnetically for 30 min. A N(CH_3_)_4_^+^-intercalated layered tetratitanate compound resulting from the calefaction of the above hydrothermal vessels with the white solution in a homogeneous reactor (KLJX-8A, Yantai Science and Technology Chemical Equipment Co. Ltd., Yantai, Shandong, China) at 100 °C for 24 h under stirring conditions. A stable white [Ti_4_O_9_]^2−^ colloidal solution was obtained by dispersing the resulting white N(CH_3_)_4_^+^-intercalated layered tetratitanate compound into 200 mL of deionized water and oscillating at room temperature for 5 days on a circular digital oscillator (SK-0330-Pro, Dalong Xingchuang Experimental Instrument Co. Ltd., Beijing, China).2.3. Synthesis of TiO_2_/CdS Composites

The 0.05 g CdS nanoparticles prepared above were transferred to four hydrothermal vessels, which containing 5.0 mL of hydrofluoric acid (HF), 2.0009 g NH_4_F, 2.0074 g urea, and 5.0 mL of formic acid (FMA), respectively. Then, 70 mL of [Ti_4_O_9_]^2−^ colloidal solution was added to the four hydrothermal vessels and stirred magnetically for 30 min. Subsequently, the four hydrothermal vessels were sealed and maintained at 180 °C for 24 h to yield a yellow solid consisting of TiO_2_ and CdS. Finally, the yellow TiO_2_/CdS composites were centrifugated and repeatedly washed with deionized water, and the corresponding TiO_2_/CdS composites were named as HF-TiO_2_/CdS, NH_4_F-TiO_2_/CdS, Urea-TiO_2_/CdS, and FMA-TiO_2_/CdS, respectively. The transformation of the [Ti_4_O_9_]^2−^ colloidal solution containing CdS and various morphology-controlling agents to TiO_2_/CdS nanocomposites in the acidic (HF and formic acid) and alkaline conditions (NH_4_F, urea), can be described as follows [33,34]:HF = H^+^ + F^−^(1)
NH_4_F = NH_4_^+^ + F^−^(2)
HCOOH = H^+^ + HCOO^−^(3)
CO(NH_2_)_2_ + H_2_O = 2NH_3_ + CO_2_(4)
Ti_4_O_9_[N(CH_3_)_4_]_2_ = [Ti_4_O_9_]^2−^ + 2[N(CH_3_)_4_]^+^(5)
[Ti_4_O_9_]^2−^ + 2H^+^ = 4TiO_2_ + H_2_O(6)
[Ti_4_O_9_]^2−^ + H_2_O = 4TiO_2_ + 2OH^−^(7)
TiO_2_ + CdS = TiO_2_/CdS(8)

The schematic of the possible growth mechanism of TiO_2_/CdS nanocomposites is depicted in Figure 1.

### 2.3. Characterization

The structure, morphology, specific surface areas, optical properties, and carrier migration and recombination of the samples were investigated using powder X-ray diffraction (XRD, XRD-6100, Shimadzu, Kyoto, Japan), field emission scanning electron microscope (FESEM, Regulus-8100, Tokyo, Japan), energy dispersive X-ray spectroscopy (EDX, Hitachi Limited, Tokyo, Japan), transmission electron microscopy (TEM, FEI TALOS 200S, Portland, Oregon, America), high-resolution TEM (HRTEM, FEI TALOS 200S, Portland, Oregon, America), specific surface area and porosity analyzer (micromeritics ASAP 2020, Atlanta, GA, USA), fluorescence spectrometer (PL, HORIBA Fluoromax-4, Kyoto, Japan), and electrochemical impedance spectroscopy (EIS, CHI600E, Shanghai Chenhua Instrument Co. Ltd., Shanghai, China); moreover, ultraviolet-visible spectrophotometer (TU-1901, Beijing Purkinje General Instrument Co. Ltd., Beijing, China) was employed to study the obtained TiO_2_/CdS composites from a photocatalytic point of view.

### 2.4. Measurement of Photocatalytic Activity

The photocatalytic performances of the as-synthesized TiO_2_/CdS nanocomposites were determined by detecting degradation of methyl orange (MO) under a 175 W low-pressure mercury lamp irradiation (Shanghai Mingyao Glass Hardware Tool Factory, Shanghai, China). Typically, 0.15 g of TiO_2_/CdS nanocomposite was suspended in 150 mL MO aqueous solution (10 ppm), and a 2 h of dark-reaction was carried out under stirring conditions to achieve the adsorption–desorption. Following, 5 mL of suspension together with the catalyst was drawn out every 15 min and centrifuged at 2500 rpm for 10 min to wipe off the TiO_2_/CdS nanocomposites. The concentrations of the MO solution were determined spectrophotometrically using a TU-1901 ultraviolet-visible spectrophotometer at a maximum absorption wavelength of 465 nm. For comparison, the photocatalytic activities of the commercial TiO_2_ (CM-TiO_2_, 96.8% anatase and 3.2% rutile) and as-synthesized CdS samples were also investigated. The stability and recyclability of the TiO_2_/CdS nanocomposites were investigated by degrading 10 ppm MO solution (150 mL). The photocatalytic degradation efficiency (*η*) can be calculated according to the formula: *η*=(1-*c*_t_/*c*_0_) × 100%, where *c*_0_ and *c*_t_ are the initial concentration of MO solution and the final concentration after irradiation for a certain time, respectively.

## 3. Results and Discussion

### 3.1. XRD Analysis

Figure 2a displays the XRD patterns of [Ti_4_O_9_]^2−^ colloidal solution, the peaks located at 9.82°and 14.98° are indexed to (400) and (600) diffraction peaks of [Ti_4_O_9_]^2−^ colloidal solution [32]. In Figure 2b–f, XRD patterns show the crystal structures of CdS and TiO_2_/CdS with 24 h of hydrothermal treatment. The sharp and strong diffraction peaks indicated that CdS and TiO_2_/CdS samples were well crystallized. Among them, the diffraction peaks appeared at 2*θ* values of 25.04°, 26.62°, 28.36°, 43.94°, 48.08°, and 52.04° can be assigned to the (100), (002), (101), (110), (103), and (112) crystal facets, respectively, of hexagonal phase CdS (JCPDS no. 65-3414) (Figure 2b). After hydrothermal treatment of [Ti_4_O_9_]^2−^ colloidal solution containing CdS (Figure 2c–f), except the diffraction peaks of CdS, 11 new diffraction peaks of anatase TiO_2_ (JCPDS no. 21-1272) were also detected at about 2*θ* values of 25.28° (101), 37.06° (103), 37.84° (004), 38.57° (112), 47.98° (200), 54.06° (105), 55.01° (211), 62.70° (204), 68.96° (116), 70.02° (220), and 75.04° (215), respectively, indicating that TiO_2_/CdS nanocomposites were successfully synthesized. The crystallite size of CdS, HF-TiO_2_/CdS, NH_4_F-TiO_2_/CdS, FMA-TiO_2_/CdS, and Urea-TiO_2_/CdS was estimated using Scherrer’s formula [35], and the average grain size was 37.6 nm, 48.1 nm, 79.2 nm, 74.5 nm, 77.8 nm. The sharp (004) diffraction peak in the TiO_2_/CdS nanocomposites represents the preferential growth of anatase {001} facets [36]. Compare with HF-TiO_2_/CdS nanocomposites, the (100) diffraction peak of CdS in the NH_4_F-TiO_2_/CdS, FMA-TiO_2_/CdS, and Urea-TiO_2_/CdS nanocomposites is not obvious, because it overlaps greatly with the (101) diffraction peak of anatase TiO_2_ in the TiO_2_/CdS nanocomposites [21].

### 3.2. FESEM and FESEM-EDS Analysis

The morphology of the K_2_Ti_4_O_9_, H_2_Ti_4_O_9_, CdS, and TiO_2_/CdS nanocomposites synthesized in the presence of different morphology control agents were characterized by FESEM. Figure 3a shows the FESEM image of K_2_Ti_4_O_9_, which composed of square-rods with the length of 0.60–2.06 μm, width of 0.13–0.26 μm, and thickness of 0.05–0.10 μm. After ion exchange, the resulting H_2_Ti_4_O_9_ remains the square-rod morphology of the precursor K_2_Ti_4_O_9_, and extends along the [10]-direction (Figure 3b) [37]. Figure 3c and d are the FESEM images of CdS, which are composed of dendritic-like dimers formed by agglomeration of nanobipyramids, nanospindles, nanocuboids, and irregular nanoparticles. Figure 3e–l displays the FESEM of TiO_2_/CdS nanocomposites obtained in the presence of different morphology control agents. For HF-TiO_2_/CdS nanocomposites, the main particle morphologies are nanobipyramids, nanocuboids and irregular nanoparticles with an average particle size of approximately 150 nm, as shown in Figure 3e,f. Comparing with Figure 3c, some pinecone-like dimers can also be observed in Figure 3e, indicating that the CdS does exist in HF-TiO_2_/CdS nanocomposites. For NH_4_F-TiO_2_/CdS nanocomposites, the main particle morphologies are nanobipyramids with the length of 90–200 nm and the width of 60–170 nm, and nanocuboids with the length of 50–170 nm and the width of 50–170 nm, as shown in Figure 3g,h. The main particle morphologies of FMA-TiO_2_/CdS nanocomposites are composed of nanobipyramids with the length of 35–77 nm and the width of 16–34 nm, and nanosquare-rods with the length of 20–110 nm and the width of 13–28 nm, as shown in Figure 3i,j. For Urea-TiO_2_/CdS nanocomposites, the main particles morphologies are nanocuboids with the length of 110–180 nm and the width of 90–160 nm, nanobipyramids with the length of 120–310 nm and the width of 100–210 nm, and truncated nanobipyramids with the length of 120–350 nm and the width of 110–180 nm, as shown in Figure 3k,l.

Energy-dispersive X-ray spectroscopy (EDX) analysis of prepared K_2_Ti_4_O_9_, H_2_Ti_4_O_9_ and CdS samples is shown in Table 1. It can be seen that the prepared K_2_Ti_4_O_9_ sample contains not only potassium, titanium, and oxygen elements, but also a small amount of carbon from the reaction raw material K_2_CO_3_. The obtained H_2_Ti_4_O_9_ sample after ion-exchange with HNO_3_ solution contains not only titanium and oxygen elements, but also a small amount of potassium and carbon from the reaction material K_2_Ti_4_O_9_ and HNO_3_, respectively, indicating that K^+^ ions have not completely replaced by H^+^ ions after three times of ion exchanges. The obtained CdS sample contains sulfur, cadmium, carbon, nitrogen, and oxygen elements, among which a small amount of carbon, nitrogen, and oxygen came from thiourea. The results of EDX analysis of TiO_2_/CdS composites prepared by hydrothermal treatment of tetra titanate colloidal suspension under the action of different morphology control agents are listed in Table 2. It can be seen that the obtained TiO_2_/CdS composites contain not only titanium, oxygen, cadmium, and sulfur elements, but also a small amount of carbon (came from CdS and N(CH_3_)_4_OH), nitrogen (came from N(CH_3_)_4_OH), and fluorine (came from HF and NH_4_F) elements. And the content of CdS particles in the prepared TiO_2_/CdS composites is different, the mass fraction (or atom fraction) of CdS in the TiO_2_/CdS composites increases in an order of HF-TiO_2_/CdS (71.583%) > FMA-TiO_2_/CdS (23.042%) > Urea-TiO_2_/CdS (22.512%) > NH_4_F-TiO_2_/CdS (21.690%). The content of CdS particles in the prepared HF-TiO_2_/CdS composite is much higher than those of TiO_2_/CdS composites, which may be due to the strong corrosivity and solubility of HF, resulting in the reduction of TiO_2_ content in the HF-TiO_2_/CdS composite [26].

### 3.3. TEM and HRTEM Analysis

The morphological and structural characteristics of the as-obtained CdS and TiO_2_/CdS samples were further investigated by TEM and HRTEM images, as shown in Figure 4 and Figure 5. A low-resolution TEM image of the as-obtained CdS sample is displayed in Figure 4a, from which a dendritic-like dimer formed by nanospindles with the width of 36–77 nm and the length of 70–230 nm is observed. The spacing of lattice fringes calculated is about 0.336 nm (Figure 4b), which can be well indexed to (101) plane of the hexagonal type CdS crystal (JCPDS no. 65-3414). Figure 4c–e shows the TEM and HRTEM images of the obtained HF-TiO_2_/CdS nanocomposite. As can be seen from Figure 4c, the HF-TiO_2_/CdS nanocomposite is mainly composed of nanocuboids with the size of 43–170 nm and nanorods with approximately 260–380 nm in length and about 85 nm in width. From Figure 4d,e, the evident and well-ordered lattice fringes with interplanar distances of 0.349, 0.349 and 0.372 nm and angles of 68.3°, 68.3° and 34.3° can be observed, corresponding to the (101), (10−1) and (002) crystal planes of the tetragonal anatase TiO_2_ phase, respectively. Therefore, the exposed crystal plane of the nanorod is {010} facets. The NH_4_F-TiO_2_/CdS nanocomposite is mainly composed of nanocuboids with the size of 60–160 nm, as shown in Figure 4f,g. The equal spacings of the lattice fringes are found to be approximately 0.349 nm with an angle of 82°, which matches the {101} and {011} facets of anatase TiO_2_, as shown in Figure 4h,i. Based on the above analysis, it can be concluded that the zone axial of the nanocuboids is along the [111] direction, that is, the top and bottom planes of the nanocuboids are perpendicular to the [111] zone axial (expressed as [111]-facets). Similarly, the nanocuboids exposed crystal planes in Figure 3c should also be [111]-facets. That is, the obtained HF-TiO_2_/CdS nanocomposite co-exposed {010} and [111]-facets.

Figure 5a–c shows the TEM and HRTEM images of the obtained FMA-TiO_2_/CdS nanocomposite. As can be seen from Figure 5a, the FMA-TiO_2_/CdS nanocomposite is mainly composed of nanocuboids, nanorods, and nanorhomboids. Clearly lattice fringes of the nanorod and nanorhomboid with the lattice spacing of 0.349 nm, corresponding to the {101} facet of the anatase TiO_2_, as shown in Figure 5b. The angles between {101} and {002} facets, {10−1} and {002} facets, {101} and {10−1} facets are about 68.3°, 68.3°, and 43.4°, which are in agreement with the theoretical angles of 68.3°, 68.3°, and 43.4°, respectively. Moreover, the {101} and {10−1} facets parallel to the lateral planes of the nanorhomboid, indicating that the co-exposed planes of the rhomboid are {010} and {101} facets. A low-resolution TEM image of the as-obtained Urea-TiO_2_/CdS nanocomposite is displayed in Figure 5d, from which nanocuboids with the size of 83–203 nm and truncated nanodiamond with 90–230 nm in length is observed. The HRTEM image shows that the angles between the bottom and lateral edges, lateral and lateral edges of the truncated diamond are 68.3°, 68.3° and 43.4°, respectively, matching closely to the values calculated by the Crystal Angle software between (101) and (002), (10−1) and (002), and (101) and (10−1) planes for the tetragonal anatase TiO_2_ (Figure 5e). Thus, the exposed planes of the truncated nanodiamond on the top/bottom and lateral planes are {010} facet and {101} facet, respectively. The HRTEM image confirms the fringe spacings of 0.349 and 0.349 nm, which are consistent with the *d* values of the (101) and (011) planes, respectively, of the tetragonal anatase TiO_2_, as shown in Figure 5f. Additionally, the (101) and (011) planes parallel to the lateral edges of the nanocuboids with an interfacial angle of 82°, therefore, we can confirm that the crystal planes exposed on the top (or bottom) and lateral planes are [111]-facets and {101} facets, respectively.

### 3.4. Nitrogen Adsorption-Desorption Isotherms Analysis

The nitrogen adsorption–desorption isotherms and corresponding pore size distribution curves of the CdS, HF-TiO_2_/CdS, Urea-TiO_2_/CdS, NH_4_F-TiO_2_/CdS, and FMA-TiO_2_/CdS samples are shown in Figure 6. All samples exhibit type-IV (IUPAC classification) and H3 adsorption-desorption isotherms, indicating the mesoporous structure, as shown in Figure 6a–c [38]. The corresponding pore size distribution in Figure 6d further confirms that as-prepared samples contain small mesopores, the average pore sizes increase in an order of CdS (39.7 nm) > FMA-TiO_2_/CdS (24.3 nm) > NH_4_F-TiO_2_/CdS (20.6 nm) > Urea-TiO_2_/CdS (15.5 nm) > HF-TiO_2_/CdS (21.3 nm). The specific surface areas of the samples increase in an order of FMA-TiO_2_/CdS (43.3 m^2^/g) > NH_4_F-TiO_2_/CdS (13.3 m^2^/g) > Urea-TiO_2_/CdS (8.3 m^2^/g) > CdS (7.4 m^2^/g) > HF-TiO_2_/CdS (5.9 m^2^/g). The as-prepared FMA-TiO_2_/CdS nanocomposite shows the higher specific surface area of 43.3 m^2^/g and the larger average pore size of 24.3 nm, which may be beneficial to MO adsorption and the improvement of photocatalytic activity [39].

### 3.5. Photoluminescence Analysis

Photoluminescence (PL) spectroscopy is an efficient technique to characterize the separation and recombination of photogenerated electron-hole pairs on the photocatalyst surface [40]. As can be seen from Figure 7, CdS, CM-TiO_2_, and TiO_2_/CdS all exhibit obvious emission peak near 560 nm, which may be due to the rapid recombination of photogenerated electron-hole pairs [23]. In addition, compared with CdS sample, some other emission peaks were observed at 394, 437, 466, 480, and 616 nm in the CM-TiO_2_ and TiO_2_/CdS nanocomposites. The first emission peak at 394 nm can be attributed the emission of the band-band PL process of anatase TiO_2_, the other emission peaks can be attributed the excitonic PL process at the band edge of anatase TiO_2_ [38,41]. As the PL spectra are produced by the recombination of charge carriers, the lower emission intensity meanings lower recombination and higher separation efficiency of photogenerated electrons and holes [42,43]. In other words, the lower emission intensities indicate the existence of oxygen defects in the prepared TiO_2_/CdS nanocomposites [44,45] The PL emission peak intensities of FMA-TiO_2_/CdS and Urea-TiO_2_/CdS are much weaker than pure CdS, CM-TiO_2_, NH_4_F-TiO_2_/CdS, and HF-TiO_2_/CdS samples, indicating the better separation of photogenerated electrons/holes. Remarkably, FMA-TiO_2_/CdS nanocomposite exhibits the lowest PL intensity in all samples, indicating that the synergistic effects of the maximum specific surface and the TiO_2_/CdS heterostructure are beneficial to impede the recombination of photogenerated carriers and enhance the photocatalytic performance [46].

### 3.6. Electrochemical Impedance Spectroscopy Analysis

The interfacial charge–carrier separation and transfer efficiency of the HF-TiO_2_/CdS, NH_4_F-TiO_2_/CdS, FMA-TiO_2_/CdS, and Urea-TiO_2_/CdS nanocomposites was investigated by Electrochemical impedance spectroscopy (EIS) measurements at a frequency range of 1.0 MHz to 0.1 Hz with a signal amplitude of 0.01 V. Nyquist plots of TiO_2_/CdS nanocomposites obtained under different morphology control agents were shown in Figure 8. It can be seen from the Nyquist plots that the order of the impedance arc radius of the nanocomposites is FMA-TiO_2_/CdS < Urea-TiO_2_/CdS < NH_4_F-TiO_2_/CdS < HF-TiO_2_/CdS, indicating that FMA-TiO_2_/CdS nanocomposite has a lower charge–transfer resistance and a higher charge–transfer efficiency, thus improving the photocatalytic efficiency [4,47].

### 3.7. Photocatalytic Activity

The ultraviolet-light-driven photocatalytic activities of TiO_2_/CdS composites prepared by adding different morphology capping agents in the mixed solution of H_2_Ti_4_O_9_ and CdS were measured by the degradation of the MO solution. As shown in Figure 9a, the optical absorption of MO did not change significantly in the absence of a photocatalyst, indicating the self-degradation of MO was negligible. Only 38.5% of MO could be photodegraded by HF-TiO_2_/CdS within 120 min, while the rate of photodegradation of MO by FMA-TiO_2_/CdS was 88.4% at the same time, and the degradation rate of MO increased in an order of FMA-TiO_2_/CdS (88.4%) > Urea-TiO_2_/CdS (67.1%) > CM-TiO_2_ (44.5%) > NH_4_F-TiO_2_/CdS (40.2%) > HF-TiO_2_/CdS (38.5%) = CdS (38.5%) > No catalyst (3.0%). For No catalyst, CdS, HF-TiO_2_/CdS, NH_4_F-TiO_2_/CdS, CM-TiO_2_, Urea-TiO_2_/CdS and FMA-TiO_2_/CdS photocatalysts, the corresponding reaction rate constants (*k*) were 0.0003, 0.0042, 0.0045, 0.0048, 0.0050, 0.0088, and 0.0167 min^−1^, respectively (Figure 9b). The FMA-TiO_2_/CdS composite exhibited the highest *k* value, approximately 55.7, 4.0, 3.7, 3.5, 3.3, and 1.9 times larger than those of No catalyst, CdS, HF-TiO_2_/CdS, NH_4_F-TiO_2_/CdS, CM-TiO_2_, Urea-TiO_2_/CdS samples, respectively. The degradation efficiency and *k* value of the FMA-TiO_2_/CdS composite are the highest, indicating that the FMA-TiO_2_/CdS composite has the best photocatalytic activity. The stability of the FMA-TiO_2_/CdS composite for MO degradation was measured by the cyclic experiment (Figure 9c). After three cycles, the degradation rate of the FMA-TiO_2_/CdS composite was 82.1%, decreased by 6.3%, indicating that the FMA-TiO_2_/CdS composite was stable.

It is widely accepted that the crystal structure, specific surface area, separation efficiency of photogenerated electron–hole pairs and exposed facets play an important role in the photocatalytic reaction [48,49,50]: the suitable heterostructure, the larger specific surface area, the higher separation efficiency of photogenerated electron–hole pairs, and the higher reactive exposed facet are conductive to the improvement of the photocatalytic activity. A possible photocatalytic mechanism of TiO_2_/CdS heterostructure for the photodegradation of MO under ultraviolet light irradiation is shown in Figure 10. Under the ultraviolet light irradiation, the photogenerated electrons (e^−^) on the conduction band (CB) of CdS can easily transfer to the CB of TiO_2_ through the heterostructure interface owing to the CB potential of CdS (−0.89 eV) is more negative than that of TiO_2_ (−0.30 eV), in contrast, the photogenerated holes (h^+^) on the valence band (VB) of TiO_2_ can migrate to the VB of CdS through the heterostructure interface owing to the VB potential of TiO_2_ (+2.90 eV) more positive than that of CdS (+1.45 eV) [5,40]. The separation of the photogenerated e^−^ (in TiO_2_) and photogenerated h^+^ (in CdS) decreases the recombination of photogenerated e^−^/h^+^ pairs, thus enhancing the photocatalytic activity of TiO_2_/CdS nanocomposites. Consequently, the photogenerated h^+^ on the VB of CdS can directly oxidize the water molecules absorbed on its surface to generate ·OH radicals, meanwhile, the photogenerated e^−^ on the CB of TiO_2_ can directly reduce the oxygen molecules absorbed on its surface to form ·O_2_ˉ radicals. The generated ·OH and ·O_2_ˉ radicals have strong oxidability, which can oxidize MO into water, carbon dioxide, and mineral acid. The possible photocatalytic degradation mechanisms are described as follows [40]:TiO_2_/CdS + *hv* → TiO_2_ (e^−^) + CdS (h^+^)(9)
TiO_2_ (e^−^) + O_2_ → TiO_2_ (·O_2_ˉ)(10)
TiO_2_ (·O_2_ˉ) + MO → TiO_2_ (CO_2_ + H_2_O + mineral acid)(11)
CdS (h^+^) + H_2_O → CdS (·OH + H^+^)(12)
CdS (H^+^)+ OH- → CdS (H_2_O)(13)
CdS (h^+^) + OH- → CdS (·OH)(14)
CdS (·OH) + MO → CdS (CO_2_ + H_2_O + mineral acid)(15)

The photocatalytic performance of the FMA-TiO_2_/CdS nanocomposite is significantly improved due to the suitable heterostructure between TiO_2_ and CdS, the larger specific area, and the higher separation efficiency of photogenerated electron–hole pairs. By comparing the TiO_2_/CdS nanocomposites, we can find that the MO degradation efficiencies for FMA-TiO_2_/CdS reached 88.4%, which was much higher than other TiO_2_/CdS nanocomposites under the same test environment conditions. The difference in photocatalytic activity among the four TiO_2_/CdS nanocomposites can be ascribed to the different specific surface areas and separation efficiency of photogenerated electron–hole pairs. The specific surface area of FMA-TiO_2_/CdS nanocomposite is much higher than those of other TiO_2_/CdS nanocomposites, which is conducive to adsorbing more MO molecules on the surface of TiO_2_/CdS nanocomposite, thereby improving the photocatalytic performance [51]. The PL intensity and the impedance arc radius of FMA-TiO_2_/CdS nanocomposite are much lower than those of other TiO_2_/CdS nanocomposites, which are beneficial to decrease the recombination of the photogenerated charge carriers and improve the charge transfer efficiency. Additionally, a large proportion of exposed {010} facets in FMA-TiO_2_/CdS nanocomposite have superior surface atom structure and surface electronic structure, which also contributes to the improvement of photocatalytic performance [52].

## 4. Conclusions

In summary, TiO_2_/CdS nanocomposites with co-exposed highly reactive anatase {101}/[111]-facets, {101}/{010} facets and {101}/{010}/[111]-facets were synthesized using different morphology control agents by a simple hydrothermal treatment of [Ti_4_O_9_]^2−^ colloidal solution containing CdS crystals. The crystal structure, morphology, specific surface area, pore size distribution, separation, and recombination of photogenerated electrons/holes of the TiO_2_/CdS nanocomposites were characterized by XRD, FESEM, specific surface area, and porosity analyzer, PL and EIS. Photocatalytic degradation of MO performance of the as-obtained TiO_2_/CdS nanocomposites was investigated under ultraviolet irradiation. It is worth mentioning that the FMA-TiO_2_/CdS nanocomposite exhibits better photocatalytic activity due to the synergistic effects of its suitable heterojunction structure, the largest specific surface area and the exposed highly reactive anatase {010} facets in comparison with the pure CdS, CM-TiO_2_ and other TiO_2_/CdS nanocomposites.

## Figures and Tables

**Figure 1 molecules-26-06031-f001:**
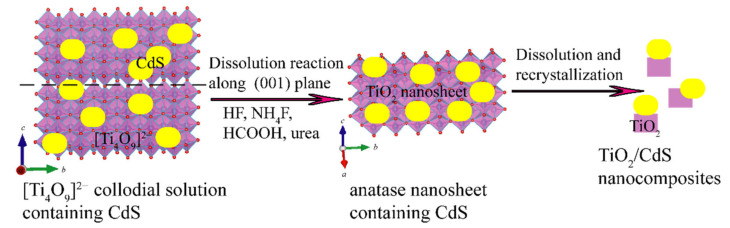
Possible growth mechanism of TiO_2_/CdS nanocomposites.

**Figure 2 molecules-26-06031-f002:**
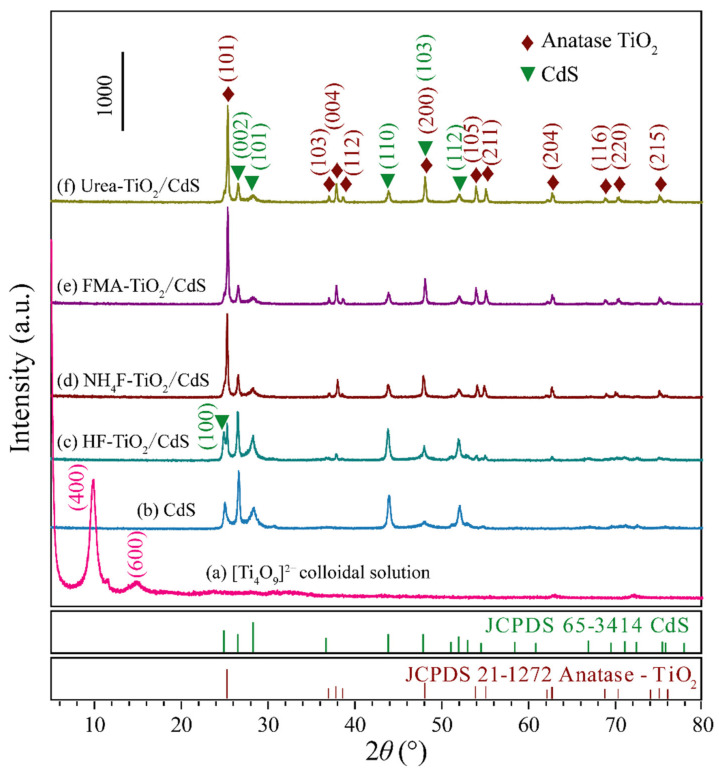
XRD patterns of as-prepared (**a**) [Ti_4_O_9_]^2−^ colloidal solution, (**b**) CdS, (**c**) HF-TiO_2_/CdS, (**d**) NH_4_F-TiO_2_/CdS, (**e**) FMA-TiO_2_/CdS, and (**f**) Urea-TiO_2_/CdS samples.

**Figure 3 molecules-26-06031-f003:**
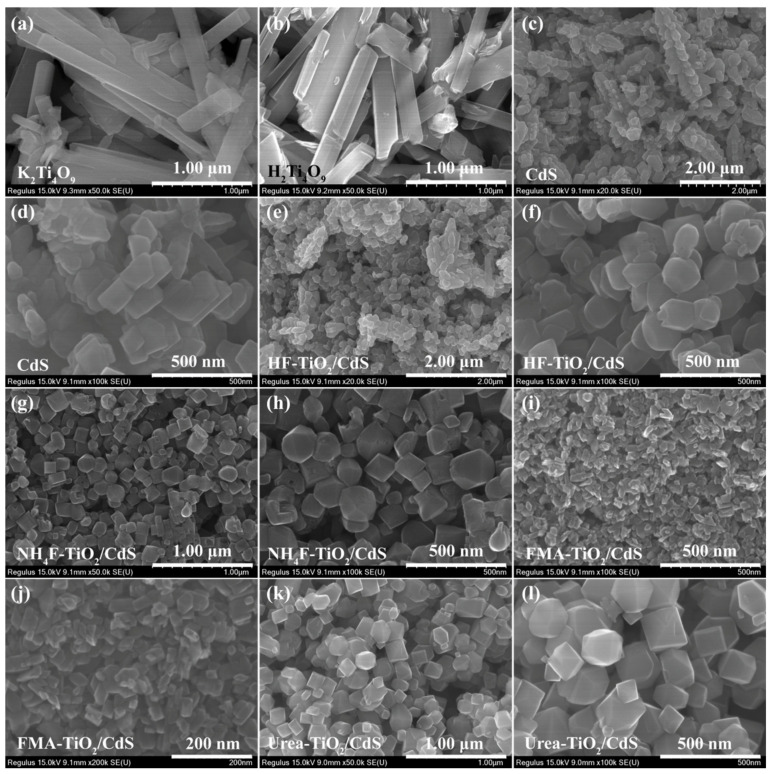
FESEM images of (**a**) K_2_Ti_4_O_9_, (**b**) H_2_Ti_4_O_9_, (**c**,**d**) CdS, (**e**,**f**) HF-TiO_2_/CdS, (**g**,**h**) NH_4_F-TiO_2_/CdS, (**i**,**j**) FMA-TiO_2_/CdS, and (**k**,**l**) Urea-TiO_2_/CdS samples.

**Figure 4 molecules-26-06031-f004:**
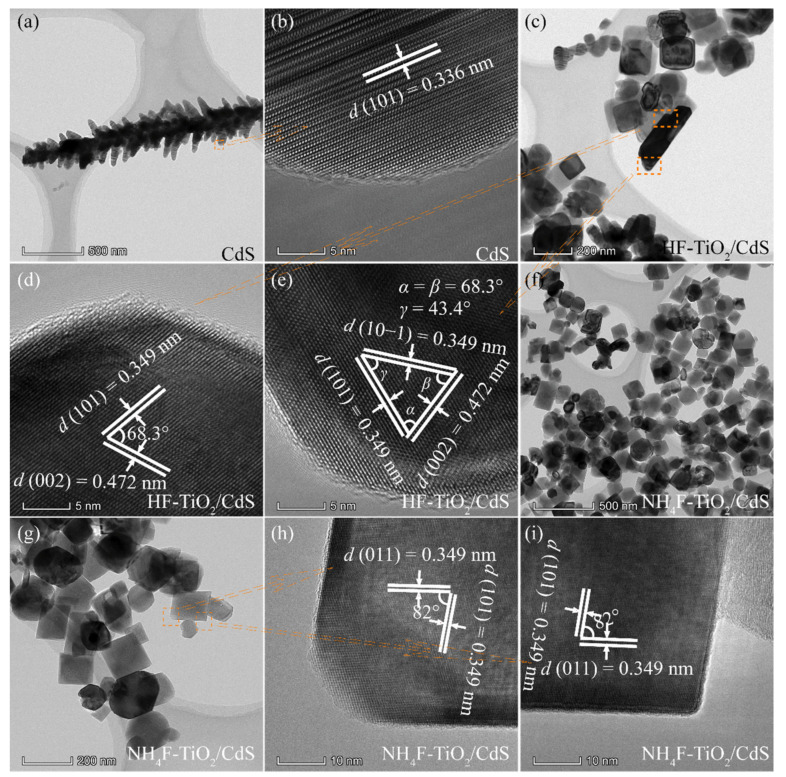
TEM and HRTEM images of (**a**,**b**) CdS, (**c**–**e**) HF-TiO_2_/CdS, and (**f**–**i**) NH_4_F-TiO_2_/CdS samples.

**Figure 5 molecules-26-06031-f005:**
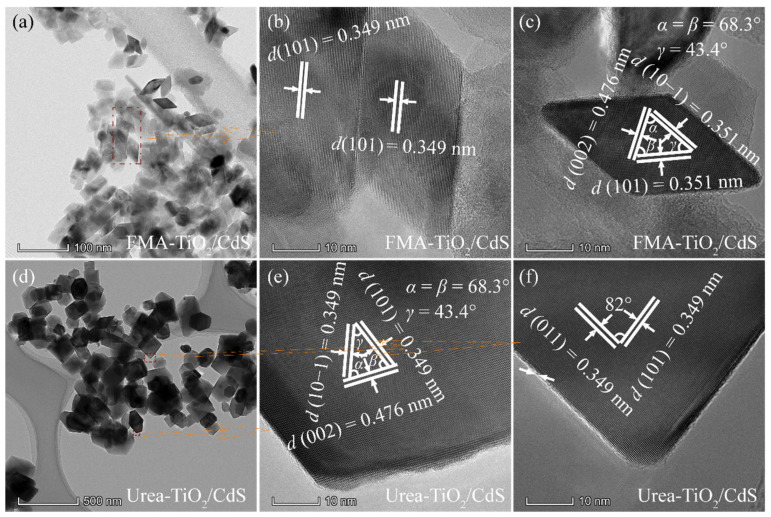
TEM and HRTEM images of (**a**–**c**) FMA-TiO_2_/CdS, and (**d**–**f**) Urea-TiO_2_/CdS samples.

**Figure 6 molecules-26-06031-f006:**
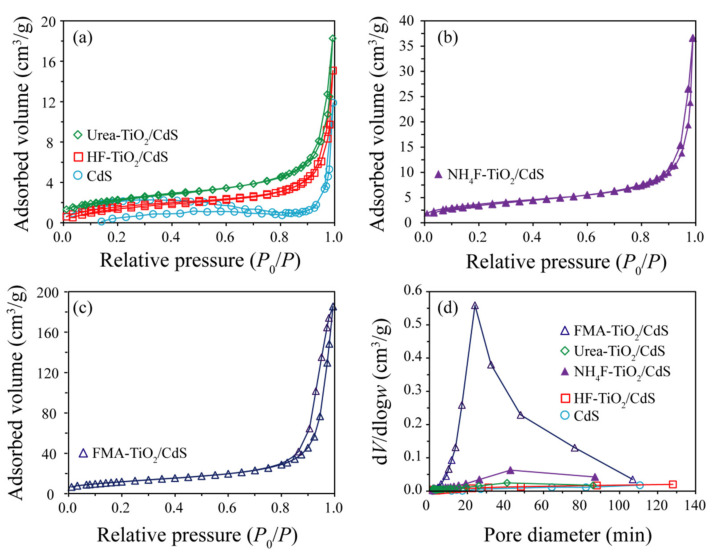
Nitrogen adsorption–desorption isotherms of the as-prepared (**a**) CdS, HF-TiO_2_/CdS, Urea-TiO_2_/CdS, (**b**) NH_4_F-TiO_2_/CdS, and (**c**) FMA-TiO_2_/CdS samples, and (**d**) the corresponding of pore size distribution curves.

**Figure 7 molecules-26-06031-f007:**
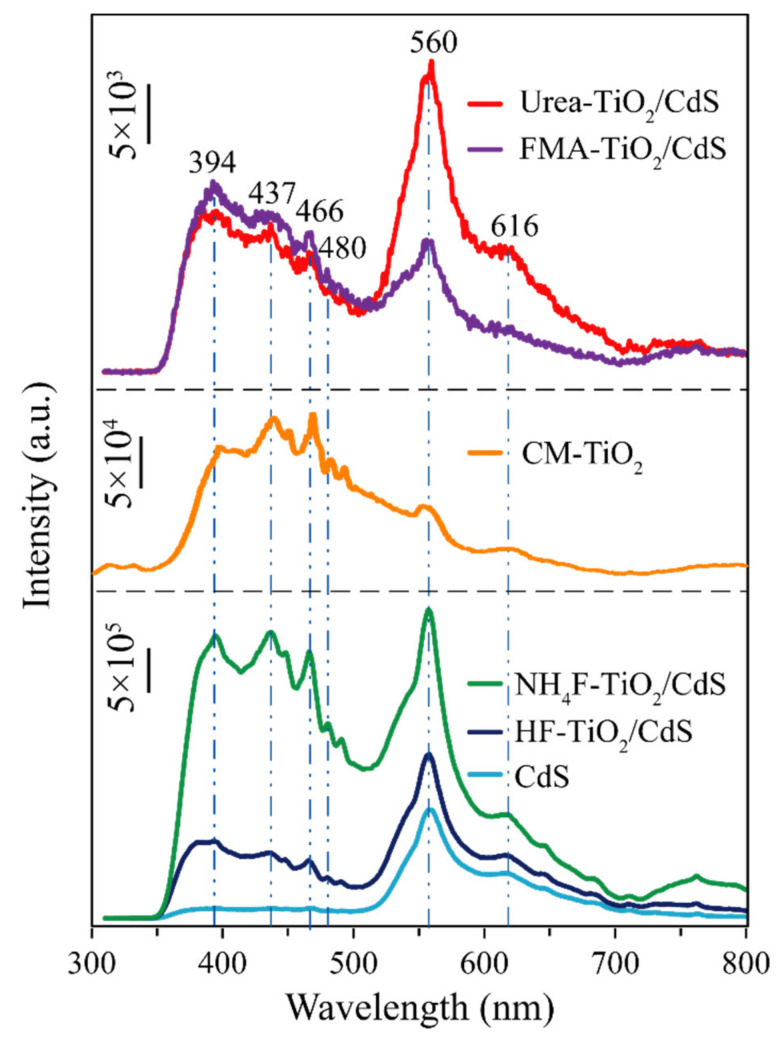
Photoluminescence spectra of the CM-TiO_2_, as-prepared CdS, HF-TiO_2_/CdS, NH_4_F-TiO_2_/CdS, FMA-TiO_2_/CdS, and Urea-TiO_2_/CdS samples.

**Figure 8 molecules-26-06031-f008:**
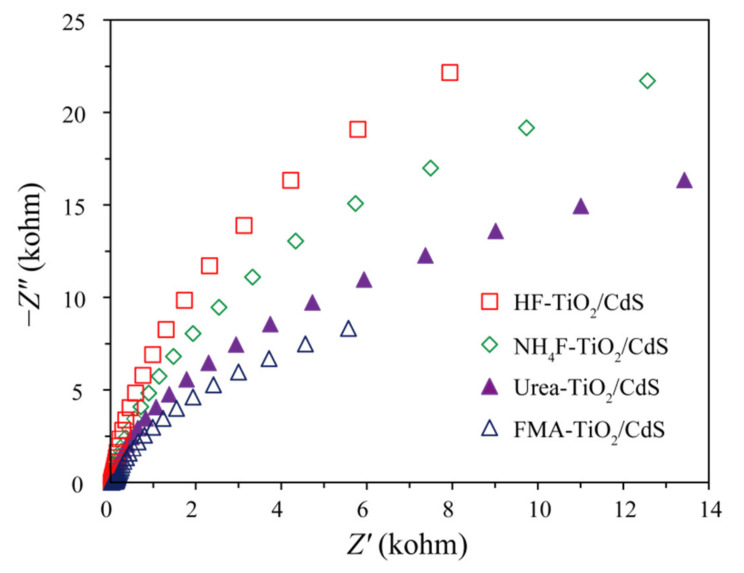
EIS Nyquist impedance plots of HF-TiO_2_/CdS, NH_4_F-TiO_2_/CdS, FMA-TiO_2_/CdS, and Urea-TiO_2_/CdS samples.

**Figure 9 molecules-26-06031-f009:**
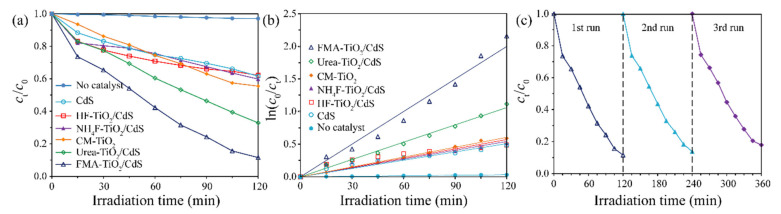
(**a**) Concentration change of the MO solution versus irradiation time and (**b**) the corresponding kinetic curves by No catalyst, CM-TiO_2_, CdS and different TiO_2_/CdS composites; (**c**) recyclability results of the photocatalytic degradation of MO by FMA-TiO_2_/CdS.

**Figure 10 molecules-26-06031-f010:**
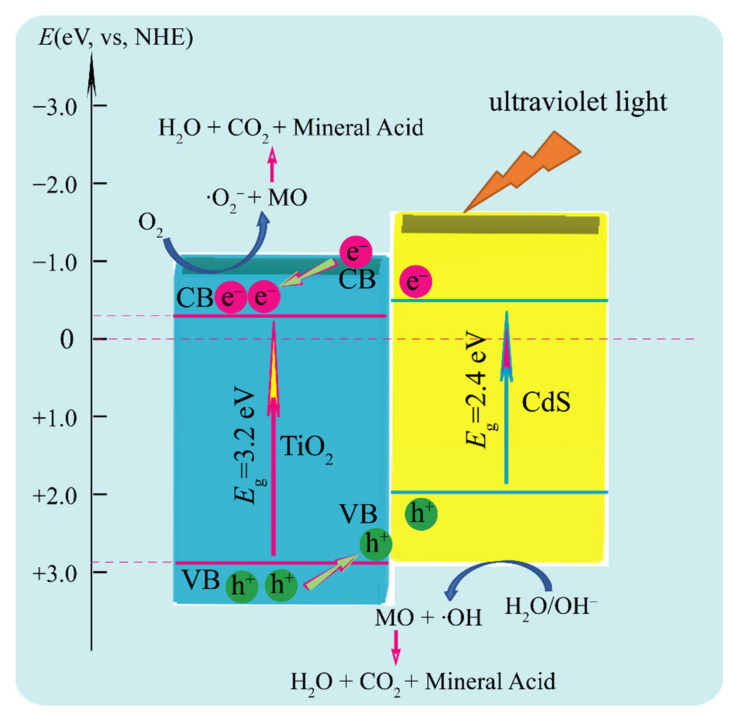
Possible photocatalytic mechanism of TiO_2_/CdS nanocomposites for the photodegradation of MO under ultraviolet light irradiation.

**Table 1 molecules-26-06031-t001:** EDX analysis of as-prepared K_2_Ti_4_O_9_, H_2_Ti_4_O_9_, and CdS samples.

Sample	K_2_Ti_4_O_9_		H_2_Ti_4_O_9_		CdS	
Element	Atom%	Mass%	Atom%	Mass%	Atom%	Mass%
C	4.869	2.019	0.000	0.000	0.889	0.136
O	49.968	27.593	47.102	23.269	0.051	0.010
K	11.328	15.286	2.027	2.448	0.000	0.000
Ti	33.835	35.103	50.005	73.908	0.000	0.000
N	0.000	0.000	0.866	0.375	1.010	0.180
S	0.000	0.000	0.000	0.000	39.849	16.286
Cd	0.000	0.000	0.000	0.000	58.202	83.387

**Table 2 molecules-26-06031-t002:** EDX analysis of different TiO_2_/CdS composites.

Sample	HF-TiO_2_/CdS		NH_4_F-TiO_2_/CdS		FMA-TiO_2_/CdS		Urea-TiO_2_/CdS	
Element	Atom%	Mass%	Atom%	Mass%	Atom%	Mass%	Atom%	Mass%
C	0.844	0.201	3.319	1.107	4.586	1.608	2.927	0.988
N	4.887	1.358	1.985	0.772	2.218	0.907	2.092	0.823
O	30.291	9.614	38.045	16.909	45.319	21.175	42.897	19.288
F	0.087	0.033	2.391	1.262	0.000	0.000	0.000	0.000
S	19.117	12.161	4.897	4.362	3.851	3.607	4.257	3.837
Ti	18.127	17.212	43.815	58.260	38.106	53.268	41.916	56.388
Cd	26.648	59.422	5.549	17.328	5.920	19.435	5.911	18.675

## Data Availability

Date are contained within the article and they are also available from the first corresponding author.

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
