# Peer review of "Synthesis and Photocatalytic Activity of TiO2/CdS Nanocomposites with Co-Exposed Anatase Highly Reactive Facets"

_molecules, 2021, doi:10.3390/molecules26196031_

Round 1

Reviewer 1 Report

The article presents a study regarding the preparation of nanocomposites based on TiO2/CdS for photocatalytic applications. The composites were investigated by XRD, SEM, TEM, EDS, PL, EIS and photocatalytic activity measurements. However, the authors must provide a better explanation for their results and a better correlation between their experimental data. For example, the authors do not provide any explanation regarding the way in which the different chemical compounds (HF, NH4F, urea, formic acid) act as structure-directing agent controlling the growth of crystals into various morphologies. In the reviewer’s opinion, the authors must improve the manuscript before publication (major revision).

Specific comments are given bellow:

  1. In the Introduction section, the authors must added a paragraph regarding the motivation for choosing HF, NH4F, urea, formic acid as structure-directing agent.
  2. Why the authors do not synthesized TiO2 in the presence of the chemical compounds (HF, NH4F, urea, formic acid) but in the absence of CdS? In this way, the authors can evidenced if CdS nanoparticles induced also modification in the TiO2 morphology during the synthesis of the composites TiO2/CdS. If studies regarding the synthesis of TiO2 only in the presence of HF, NH4F, urea, formic acid are already published, the authors must cited and referred to them in order to emphasize that CdS nanoparticles can influence the TiO2 morphology. If such studies are not published, the authors must added new experimental data regarding this important aspect.
  3. In the synthesis of TiO2/CdS composites section, the authors must provide the chemical reactions and the possible mechanism (schematic representation) involved in the synthesis of TiO2/CdS composites in the presence of structure-directing agents (HF, NH4F, urea, formic acid). Further, if they use chemical recipes from other works, they must give these references. Otherwise, they must provide more details regarding the experimental parameters. For example: Why they choose to use these particular chemical compounds HF, NH4F, urea, formic acid in the amounts specified in their work?
  4. Why the authors choose methyl orange as dye for their photocatalytic measurements? They tests other dyes such as methylene blue or rhodamine B?
  5. The authors must provide references in order to sustain their claim: “Fig. 1(a) displays the XRD patterns of [Ti4O9]2- colloidal solution, the peaks located at 135 9.82°and 14.98° are indexed to (400) and (600) diffraction peaks of [Ti4O9]2- colloidal solution.“
  6. In the EDS section, the paragraph “The atom fraction of titanium and cadmium is 18.127% and 26.648% in HF-TiO2/CdS composite, 43.815% and 5.549% in NH4F-TiO2/CdS, 38.106% and 5.920% in FMA-TiO2/CdS composite, 41.916% and 5.911% in Urea-TiO2/CdS composite, respectively. The mass fraction of titanium and cadmium is 17.212% and 59.422% in HF-TiO2/CdS composite, 58.260% and 17.328% in NH4F-TiO2/CdS composite, 53.268% and 19.435% in FMA-TiO2/CdS composite, 56.388% and 18.675% in Urea-TiO2/CdS composite, respectively” must be deleted because all these data are presented in Table 2.
  7. The authors must added an explanation for this result: “Therefore, the content of CdS particles in the prepared TiO2/CdS composites is different, the mass fraction (or atom fraction) of CdS in the TiO2/CdS composites increases in the order of HF-TiO2/CdS (71.583%) > FMA-TiO2/CdS (23.042%) > Urea-TiO2/CdS (22.512%) > NH4F-TiO2/CdS (21.690%). The content of CdS particles in the prepared HF-TiO2/CdS composite is much higher than those of TiO2/CdS composites.”
  8. The authors must investigated their composites by UV-VIS absorption spectroscopy, the absorbance data being further correlated with the photocatalytic data.
  9. PL is an investigation technique not a technology. The authors must correct this aspect.
  10. The authors must provide references in order to sustain their claim: “As can be seen from Fig. 6, CdS and TiO2/CdS have an obvious emission peak at 560 nm, which may be caused by the rapid recombination of photogenerated electron-hole pairs.” For comparison reason, the authors must added the PL spectrum of TiO2. Furthermore, taking into account that the emission peaked at 560 nm can be noticed in both CdS and TiO2/CdS it is unlikely that the origin of this emission is “the rapid recombination of photogenerated electron-hole pairs”.
  11. The authors must provide references in order to sustain their claim: “In addition, compared with CdS sample, some other emission peaks were observed at 394, 437, 466, 480, and 616 nm in the TiO2/CdS nanocomposites.” Most probably, these emission are due to the TiO2, the authors must provide and comment the origin of all these emission.
  12. In the Conclusion section, the authors must added a paragraph with the most important results provided by each investigation technique.
  13. The English must be revised in the entire manuscript.

The authors must provide a point-to-point response.

Author Response

Dear reviewer! Attached is my reply to your review comments.

Reviewer 2 Report

The paper reports on the the synthesis and photocatalytic activity study of several TiO2/CdS nanocomposites prepared by hydrothermal route. The subject of the paper is within the scope of Molecules journal.

I have the following comments:

  1. Recent papers and reviews concerning photocatalytic activity of fluorinated titania and TiO2/CdS composites are largely ignored. No comparison is provided of the results obtained with the existing reports on TiO2/F and TiO2/CdS. Please refer to doi 10.1016/j.jphotochem.2004.02.006, 10.1016/j.rser.2015.10.100, 10.1016/j.jphotochem.2015.01.010, 10.1038/nature06964, 10.1016/j.molcata.2012.01.006.
  2. Please provide the information on the sizes of both TiO2 and CdS nanoparticles.
  3. Line 188: Please replace "carbon elements" with "carbon".
  4. Please replace "Energy spectrum analysis" with "EDX analysis" (Tables 1&2).
  5. Line 259: Please revise the data presented (0.349 and 0.349 nm).
  6. Both the English language and grammar in the paper need improvement.

Author Response

(The authors gave the same response as above.)

Round 2

Reviewer 1 Report

In Figure 1, “recation” must be replaced by “reaction”. The authors responded satisfactorily to the reviewer comments.